# Sub-Millisecond Photoinduced Dynamics of Free and EL222-Bound FMN by Stimulated Raman and Visible Absorption Spectroscopies

**DOI:** 10.3390/biom13010161

**Published:** 2023-01-12

**Authors:** Yingliang Liu, Aditya S. Chaudhari, Aditi Chatterjee, Prokopis C. Andrikopoulos, Alessandra Picchiotti, Mateusz Rebarz, Miroslav Kloz, Victor A. Lorenz-Fonfria, Bohdan Schneider, Gustavo Fuertes

**Affiliations:** 1Institute of Biotechnology of the Czech Academy of Sciences, 25250 Vestec, Czech Republic; 2ELI Beamlines Facility Extreme Light Infrastructure ERIC, 25241 Dolni Brezany, Czech Republic; 3Faculty of Science, Charles University, 12800 Prague, Czech Republic; 4The Hamburg Centre for Ultrafast Imaging, Hamburg University, 22761 Hamburg, Germany; 5Institute of Molecular Science, University of Valencia, 46980 Paterna, Spain

**Keywords:** flavins, protein structural dynamics, light-oxygen-voltage (LOV) photosensors, femtosecond-stimulated Raman spectroscopy (FSRS), transient visible absorption (visTA) spectroscopy, photochemistry, photobiology, time-resolved vibrational spectroscopy, kinetic isotope effect (KIE), maximum entropy method, lifetime distribution analysis (LDA)

## Abstract

Time-resolved femtosecond-stimulated Raman spectroscopy (FSRS) provides valuable information on the structural dynamics of biomolecules. However, FSRS has been applied mainly up to the nanoseconds regime and above 700 cm^−1^, which covers only part of the spectrum of biologically relevant time scales and Raman shifts. Here we report on a broadband (~200–2200 cm^−1^) dual transient visible absorption (visTA)/FSRS set-up that can accommodate time delays from a few femtoseconds to several hundreds of microseconds after illumination with an actinic pump. The extended time scale and wavenumber range allowed us to monitor the complete excited-state dynamics of the biological chromophore flavin mononucleotide (FMN), both free in solution and embedded in two variants of the bacterial light-oxygen-voltage (LOV) photoreceptor EL222. The observed lifetimes and intermediate states (singlet, triplet, and adduct) are in agreement with previous time-resolved infrared spectroscopy experiments. Importantly, we found evidence for additional dynamical events, particularly upon analysis of the low-frequency Raman region below 1000 cm^−1^. We show that fs-to-sub-ms visTA/FSRS with a broad wavenumber range is a useful tool to characterize short-lived conformationally excited states in flavoproteins and potentially other light-responsive proteins.

## 1. Introduction

Light-triggered protein structural dynamics spans a wide range of timescales, from femtoseconds to hours [1,2]. In the light-oxygen-voltage (LOV) class of blue-light photoreceptors, photon absorption by the embedded oxidized flavin chromophore (typically flavin mononucleotide or FMN) initiates a cascade of transient events [3,4,5], eventually leading to activation of enzymatic activity or to binding/unbinding reactions with cellular partners. For instance, in the two-domain LOV/helix-turn-helix transcription factor EL222 from the bacterium *Erythrobacter litoralis*, photo-stimulation drives oligomerization, protein-DNA association, and gene expression [6,7,8]. The photoproduct is a thioether linkage between the C4a atom of FMN and the Sγ atom of a conserved cysteine (C78 in EL222) residue present in the binding pocket of LOV domains [9]. Interest in LOV photosensors has raised since they can be utilized to confer photoresponsiveness to otherwise non-light-sensing proteins. Therefore, LOV modules are becoming powerful tools in optogenetics and synthetic biology applications that require the spatial and temporal photocontrol of ion channels, gene expression, or catalysis, among others [10,11,12].

Electronic and vibrational spectroscopies have provided valuable insights on the photoactivation mechanism of LOV sensors [13]. UV/Visible and transient visible absorption (visTA) spectroscopies allow researchers to track the metastable cysteinyl-flavin adduct state of LOV domains, which is characterized by an absorption band around 390 nm and thermally decays to the ground state in a matter of seconds to hours, depending on the specific sequence and solvent conditions [14]. Time-resolved femtosecond-stimulated Raman spectroscopy (FSRS) can map chromophore and protein structural dynamics down to the femtosecond time scale [15,16,17]. Time-resolved FSRS is highly complementary to the more commonly employed time-resolved infrared (TRIR) spectroscopy for the study of light-sensitive proteins [18,19,20] and has been recently applied to investigate flavins and flavoproteins [21,22,23]. A major advantage of FSRS is the minimal background signal of liquid water (H_2_O), while for TRIR, deuterium oxide (D_2_O) is the solvent of choice, particularly when monitoring the structure-sensitive amide I band of proteins between 1700 and 1600 cm^−1^. A second benefit of FSRS is the feasibility of acquiring a large spectral range (potentially a few thousand wavenumbers from 50 cm^−1^ to 3600 cm^−1^, which is equivalent of measuring dipole active vibrations from THz to near-IR) in a single experiment [21], while TRIR spectra suffer from a narrow detection frequency window (usually a few hundred wavenumbers [24]). Yet, virtually all FSRS studies of proteins so far have concentrated on ultrafast phenomena up to the nanosecond regime [25,26,27], thus leaving relevant biological time scales unaddressed. This issue is particularly important for LOV proteins, since the primary photochemistry -adduct formation, proton and electron transfer reactions, and subsequent conformational changes all happen around microseconds [24,28,29].

Here we developed a set-up able to record simultaneously transient absorption spectra in the UV/visible/near-IR region (from ~380 to ~1200 nm) and FSRS (from ~200 to ~2200 cm^−1^) both from femtoseconds to sub-milliseconds (~0.8 ms) after photoexcitation with a tunable actinic pump. This represents a considerable upgrade to existing FSRS set ups [30,31,32,33,34]. We then monitored the photoinduced structural dynamics of FMN in three different environments: (i) free in solution, (ii) bound to wild-type EL222 (EL222-WT), and (iii) bound to a cysteine-less variant (EL222-C78A) unable to form the adduct state. For EL222-WT, we detected all three previously proposed intermediate species based on TRIR: excited singlet (1FMN*), excited triplet (3FMN*), and adduct (A390) states [24,35]. In the case of EL222-C78A, where covalent bond formation is impaired much like in free FMN, a long-lived triplet state was observed. Importantly, we found evidence of additional dynamical events. By repeating the FSRS measurements in H_2_O- and D_2_O-based buffers, we determined the kinetic isotope effect of all elementary steps in the photocycle. In combination with quantum chemistry calculations of flavins and their clusters [22,35,36,37], we expect that fs-to-ms broadband TA/FSRS experiments can yield richer information on flavoprotein structural dynamics.

## 2. Materials and Methods

FMN (sodium salt) was purchased from Cayman Chemical Company and used without further purification. EL222-WT, EL222 (17–225), was purified as previously described [38]. A Cys78-to-alanine mutant of EL222 (EL222-C78A) was generated by site-directed mutagenesis and purified as the wt version was. Protein purity was confirmed by SDS-PAGE (Appendix A). Both FMN and EL222 were prepared in 50 mM MES 100 NaCl pH = 6.8 (H_2_O-based buffer). For experiments in D_2_O, FMN was directly dissolved in 50 mM MES 100 NaCl pD = 6.8 (D_2_O-based buffer). EL222 was buffer exchanged by size-exclusion chromatography using a Superdex 75 Increase 10/300 column. Sample volumes (concentrations in parenthesis) for the FSRS experiment were ~10 mL (2 mM), ~800 µL (2 mM), and ~800 µL (1 mM) for FMN, EL222-WT, and EL222-C78A, respectively.

The femtosecond-stimulated Raman spectroscopy setup (Appendix A), based on the spectral watermark method [39], is an upgraded version of the one described in our previous work [22]. In the current design, we seeded two independent 1 kHz chirped pulse amplifiers (CPAs) with femtosecond pulses from one shared Ti: sapphire oscillator. We delayed the seed both electronically and optically to trace processes beyond six nanoseconds. To initiate photoreaction, we employed 1 μJ 475 nm from TOPAS as the actinic pump. Meanwhile, we focused the 1450 nm signal beam from a second TOPAS system on a moving CaF_2_ plate to generate a white light supercontinuum as the probe. The 800 nm femtosecond pulses from the second amplifier passed through a home-built pulse shaper to create a series of frequency-locked picosecond pulses as the Raman pump. The energy of the Raman pump was 5 μJ. We implemented 53 exponentially spaced time delays from 100 femtoseconds to 0.8 milliseconds to sample the photoinduced dynamics of flavin molecules in different surroundings. All the above experiments were taken under the magic-angle (54.7°) condition to remove the influence of orientational relaxation. To reduce the impact of photodamage in the measurement, we flushed the sample with a 6.0 mL/min (30 rpm) flow rate. In the case of EL222-C78A, the sample was purged by nitrogen gas during the measurements in order to minimize the generation of damaging reactive oxygen species, which can arise from the reactive triplet state [40,41].

Time-resolved traces (Raman intensity/absorbance change as a function of time and frequency/wavelength) were subjected to lifetime distribution analysis (LDA) by the maximum entropy method [38,42,43,44]. The time constants were fixed and logarithmically distributed from 10^−13^ to 10^−3^ s with 41 points per decade. The amplitudes $a_{i,k}$, dependent on wavenumber k and time constant $\tau_{k}$, were the only fitting parameters. The L-curve criterion was used to select the optimal regularization parameter. From the lifetime density maps, the average dynamical content D was calculated as [45,46]:$$D\left(\tau_{k}\right)=\sqrt{\underset{i}{\sum }a_{i,k}^{2}}$$

The peak center and areas of the D lifetime distribution were calculated. To avoid over-interpretation, only peaks with relative areas larger than 10% were considered significant. Four *D* lifetime distributions were calculated, one for the transient absorption data (380–1000 nm) and three for the FSRS data, truncated low frequency (303–1000 cm^−1^), full low frequency (~149–1000 cm^−1^, see exact range in Appendix A), and high frequency (1000–1750 cm^−1^). Time delays below 750 fs were excluded from the analysis.

Time-resolved FSRS datasets of FMN and EL222-WT, both in H_2_O, were also subjected to global kinetic analysis (GKA) using Glotaran software [47,48]. A three-component sequential model was applied and the evolution-associated difference spectra (EADS) of the first (singlet) and second (triplet) components were retrieved. The third spectral component had a low signal-to-noise ratio (as a consequence of being non-resonant with the Raman pump) and was omitted for clarity.

## 3. Results

### 3.1. Time-Resolved FSRS/Absorption of FMN, EL222-WT and EL222-C78A

We have characterized the non-equilibrium dynamics of FMN in aqueous solution, both free and associated to protein cages (Appendix A), from the initial ground state to the species populated at almost one millisecond after blue-light (475 nm) excitation. The transient spectra, stimulated Raman and UV/visible/near-IR absorption, as a function of time delay and frequency (or wavelength) of FMN, EL222-WT, and EL222-C78A can be found in Figure 1, Figure 2 and Figure 3, respectively. In order to gain more insight into the reaction 

Rates and reaction mechanism, all measurements have been done in H_2_O and D_2_O solvents. Negative signals (blue color) correspond to depopulated ground states, and positive signals (red color) arise from the excited state or new ground state populations.

Due to resonance enhancement, FSRS is particularly sensitive to those transient states where the absorption wavelength of an electronic transition is close to that of the Raman pump (800 nm in our case). FMN species in the electronic ground state, including the aforementioned adduct state, absorb maximally between 390 and 450 nm and are thus non-resonant with the incident wavelength. This causes a dramatic drop in Raman intensity at the longest pump-probe delays, usually beyond microseconds (see Figure 1 and Figure 2). In the case of EL222-C78A, the Raman signal does not vanish, probably due to the presence of a resonant non-decaying component i.e., a component with a lifetime similar or longer than the experimentally available time window.

### 3.2. Analysis of Time-Resolved FSRS/visTA Datasets

Global kinetic analysis (GKA) methods are often used to extract the transient spectra and associated lifetimes from time-resolved spectroscopic data [47,49]. GKA requires the assumption of a given number of components and connectivity among them. On the other hand, methods based on lifetime distribution analysis (LDA) are essentially model-free since the number of components need not be predefined [49,50]. In order not to restrict ourselves to previously described sequential models of FMN photochemistry and also to exploit the full potential of our unique datasets covering extended frequency and time ranges, we prioritized LDA methods. In a first step, datasets were converted from the time domain to the lifetime domain by inverse Laplace transform using the maximum entropy method [42,43,44]. The corresponding lifetime density maps for FMN, EL222-WT, and EL222-C78A can be found in Appendix A, respectively. In a second step, we calculated the so-called “average dynamical content” (abbreviated as *D*), which is a convenient way to visualize the time scales at which spectral changes occur [45,46]. A threshold-frequency of 1000 cm^−1^ was used to split the Raman data into high-wavenumber (1750–1000 cm^−1^) and low-wavenumber (1000–149 cm^−1^ depending on the dataset, see the actual minimum wavenumbers in Appendix A) regions. Normalized dynamical contents (*D*) as a function of lifetime are plotted in Figure 4 for wavenumbers higher than 303 cm^−1^ to compare the datasets taken in H_2_O and D_2_O in an unbiased way. *D* lifetime distributions using the whole range of Raman shifts appear in Appendix A. Mean lifetimes are listed in Table 1.

All datasets show at least two dynamical events. The fastest event lies in the low nanosecond time scale (2–4 ns, labeled “one” in Figure 4). The slowest event is in the sub-microsecond to microsecond time scale (0.3–300 μs, labeled “two” in Figure 4). The first can be assigned to intersystem crossing (ISC) between the singlet (1FMN*) and triplet (3FMN*) states. The nature of the latter event depends on the particular environment of the FMN. For FMN bound to EL222-WT, the most obvious assignment would be the formation of FMN-cysteinyl adduct (A390) state that in turn necessitates ISC from triplet to ground states. In the case of EL222-C78A variant, incapable of covalent bond formation, the transition may correspond to the relaxation of FMN triplet state back to the ground state by phosphorescence emission [51,52]. The interpretation of the second dynamical event of FMN (sub-microseconds) is not as straightforward although photon re-emission by phosphorescence is unlikely since it typically happens on the microsecond (and longer) time scales.

## 4. Discussion

Despite many decades of intense research, a detailed understanding of LOV photochemistry is missing [4]. Intermediate species in the microsecond/millisecond time scales have been postulated but their experimental identification and characterization remains elusive [53]. The long-standing view that certain protein residues are essential for signal transduction, such as the conserved cysteine and glutamine residues, has been challenged recently [54,55,56]. In principle, new methodological developments could fill in this knowledge gap. Here we show that a broad time scale (~50 fs to ~1 ms) and frequency range (380–1200 nm and 200–2200 cm^−1^) dual transient visible absorption/FSRS set-up in combination with lifetime distribution analysis reveals at least two dynamical events experienced by FMN, free and EL222-bound, upon photoexcitation.

To get further insights into the resolved dynamic events we calculated the kinetic isotope effect (KIE) caused by deuteration of labile protons, determined as the ratio between the lifetimes when using D_2_O and H_2_O solvents (Appendix A). The value of KIE helps to identify if breaking X-H (where X = O, N, or S) bonds is rate-limiting for a given dynamic step, such as proton transfer reactions [57]. A KIE ~ 1 was found for the singlet-to-triplet state transition of FMN, irrespective of its surroundings (free or associated with EL222), in agreement with the behavior of EL222-WT and other LOV proteins measured by visible transient absorption and infrared spectroscopies [24,58,59]. This result suggests that in the singlet-to-triplet transition proton transfer reactions are absent (or if they take place they are not rate-limiting). In contrast, the triplet-to-adduct conversion of EL222-WT featured a large isotope effect, with a KIE of ~7–12, depending on the probed frequency region. A similar large KIE has been observed in EL222-WT and other LOV proteins using TRIR spectroscopy [24,60], and indicates a proton transfer reaction as a rate-limiting step in this transition. The sub-millisecond decay of EL222-C78A, unable to form an adduct, showed virtually no dependence on deuteration, indicating that this mutation prevented the proton transfer reaction taking place in the WT form. Finally, the second dynamical event for free FMN showed a large KIE (~4–9) in spite of being unable to form an adduct. Such a large KIE suggest that this dynamical event is unlikely to originate from phosphorescence, but could be due to the transient formation of FMN radicals [61,62], as further discussed below.

Apart from these two highly abundant transitions, LDA indicates the existence of additional dynamical events in several datasets (labeled with Greek letters in Figure 4 and Appendix A). Electronic transitions in the UV/visible/near-IR range witness more than two dynamical events, especially in the case of free FMN. *D* lifetime distributions of FMN in H_2_O (Figure 4A) suggests the presence of a sub-nanosecond (α) and a sub-millisecond (β) dynamical events. Similarly, the photodynamics of FMN in D_2_O features two transitions (γ and δ) preceding the equilibration of the singlet state (Figure 4B). The integrated spectra corresponding to these additional events are clearly distinct (Appendix A) suggesting that they indeed belong to different species. Even more components can be resolved in one dataset (FMN in D_2_O) by including wavenumbers below 303 cm^−1^ (see peaks ε, ζ, η, and θ in Appendix A and their associated spectra in Appendix A). Some of these components can be interpreted based on the timescales at which they occur (Table 1). For instance, the two dynamical events happening in the ~1 ps time regime (γ and ε) can be attributed to the delayed solvent response to FMN vertical excitation [22]. The two components in the ~100 ps time scale (δ and ζ), which are again only observed in free FMN, may be related to the folding of the ribityl chain against the isoalloxazine ring [22], a process that cannot occur in protein-bound FMN.

Although a definitive assignment of the observed spectral bands linked to these dynamical events is beyond the scope of the present work, some interpretations are possible. In our recent article, we exhaustively analyzed the impact of deuteration on the vibrations of EL222 during the main dynamical events [35]. While the bulk of that work covered vibrations in the fingerprint region above 1000 cm^−1^, these newly reported events manifest in the low-frequency region, which is populated by flavin out-of-plane (OOP) C-H and ring bending modes. In particular, the peaks around 370 cm^−1^ prominent in the spectra of the first major transition seen in EL222-WT (Appendix A), could be assigned to the aforementioned OOP ring modes coupled with modes of the proximal cysteine, which is reinforced by the absence of these bands in the EL222-C78A mutant (Appendix A). These low-frequency dynamical events warrant further extensive analyses, similar to the ones documented for events pertaining to the Raman high-frequency region [22,23,36]. Regarding the β component, the positive band at ~570 nm resembles that observed in transient spectra of other flavoproteins, which has been assigned to the formation of neutral semiquinone radicals (FMNH•) [56,63].

We then compared the *D* lifetime distributions of EL222-WT in D_2_O obtained from time-resolved FSRS (this work) and infrared spectroscopies (reanalysis of the data presented in our previous work [35]) using a common frequency range (1500–1750 cm^−1^, Appendix A). The mean lifetimes are essentially the same, indicating that both vibrational techniques can provide accurate values of the timescales underlying photobiochemical processes. However, as mentioned before, the wider frequency range in Raman experiments increases the information content and gives access to more dynamical events.

Our transient spectra of FMN in H_2_O obtained from GKA are in excellent agreement, in the commonly used 1100–1700 cm^−1^ region with previously published transient spectra by Iuliano et al. [23] (Appendix A). A notable exception is the band at 1573 cm^−1^ assigned to FMN carbonyl stretches [22], which was absent in the previous study. Moreover, the transient spectra of EL222-WT are similar to that of AsLOV2 [23], a widely studied LOV-containing protein from the plant *Avena sativa* (Appendix A). The transient spectra corresponding to the singlet species features essentially the same bands in both proteins, although the relative intensities of each peak tend to differ between the two (Appendix A). The transient spectra corresponding to the triplet intermediate are virtually identical for EL222-WT and AsLOV2 (Appendix A). Since the residues present in the binding pocket of both proteins are partially different (Appendix A), these results suggest that resonance enhancement affects primarily chromophore-localized modes, as previously reported [23]. The reactive cysteine does not have a clear impact on the Raman spectra since the singlet and triplet states of EL222-C78A (Appendix A) are nearly identical to those arising from EL2222-WT (Appendix A). However, residual contributions from H-bonding residues (particularly asparagines) cannot be completely ruled out and may help explaining the subtle differences in frequency and/or intensity between FMN bound to EL222-WT, EL222-C78A and AsLOV2, particularly for the excited singlet state.

In those instances where only two main dynamical events are present, e.g., EL222-WT in H_2_O, the results from GKA and LDA are in quantitative agreement (Appendix A). However, when photoinduced dynamics are more complex, LDA analysis provides richer information by disclosing dynamical events that are otherwise silent in GKA.

## 5. Conclusions

We have shown that a hybrid set-up able to simultaneously record transient visible spectra and femtosecond-stimulated Raman spectra up to nearly one millisecond after photoexcitation can provide detailed kinetic information of free and protein-bound flavins. Critically, *D* lifetime distributions in the low-frequency Raman region tend to be more convoluted than in all other probed frequency regions (Raman high-frequency, infrared high-frequency and near-UV/visible/near-infrared) possibly indicating the presence of additional short-lived states beyond the well-known single, triplet, and adduct species. Additionally, the flexibility of Raman spectroscopy in terms of solvent choice, allowed us to compare the kinetics in H_2_O and D_2_O and thus to determine the role of proton transfer in different stages of the LOV photocycle.

Both chromophore and polypeptide modes contribute to the infrared spectra, which in turn provide a more holistic view of the protein under investigation. On the other hand, the transient Raman spectra of LOV proteins using 800 nm Raman excitation reports almost exclusively on excited states of the FMN moiety itself. However, it seems that the excited singlet state is more sensitive to residue alterations in the FMN binding pocket than the excited triplet state. Because of sequence and fold conservation across LOV photosensors [64], the spectra corresponding to their photoexcited singlet and triplet states are expected to be rather similar, at least in the high-frequency region. Nevertheless, how exactly the local protein environment influences the Raman spectra of LOV proteins remains an open question.

Other degrees of selectivity may be attained by using distinct excitation wavelengths. For instance, tuning the Raman pump to ~400 nm would make the adduct state resonant and thus “visible”. Raman pumps in the UV region i.e., ultraviolet-resonance Raman (UVRR), would excite the aromatic side-chains or the protein backbone depending on the chosen wavelength, thereby providing more residue-specific information [65].

Finally, yet importantly, the approach described here may be useful to interrogate photoinduced dynamics in other light-responsive biomolecules.

## Figures and Tables

**Figure 1 biomolecules-13-00161-f001:**
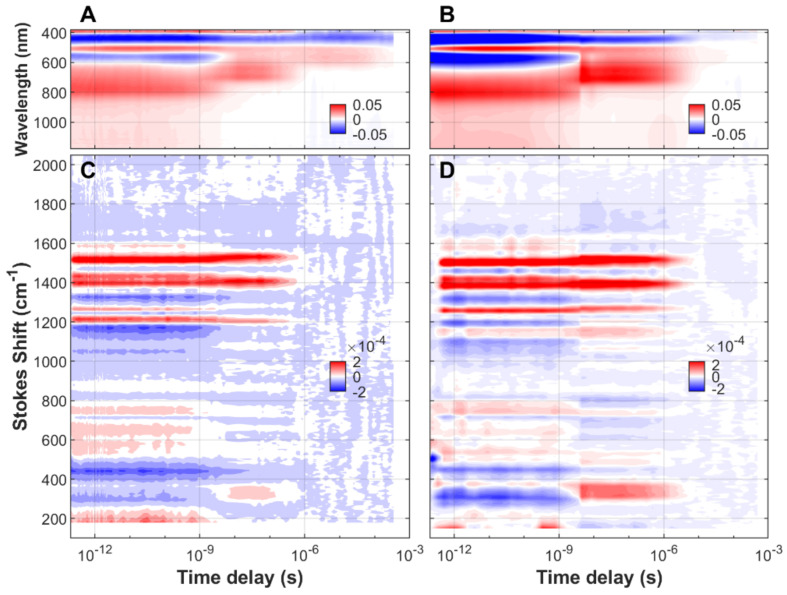
Time-resolved sup-picosecond to sub-millisecond Raman/absorption spectra of free FMN upon excitation with a 475 nm actinic pump. (**A**) TA spectroscopy in H_2_O-based buffer. (**B**) TA spectroscopy in D_2_O-based buffer. (**C**) Transient FSRS in H_2_O-based buffer. (**D**) Transient FSRS in D_2_O-based buffer. The buffer was MES 50 mM NaCl 100 mM at pH (or pD) = 6.8. The Raman pump was set to 800 nm. The red color indicates a positive change of Raman intensity (FSRS contour plots) or a positive absorbance change (TA contour plots). On the contrary, the blue color code indicates a negative change of Raman intensity (FSRS) or a negative absorbance change (TA).

**Figure 2 biomolecules-13-00161-f002:**
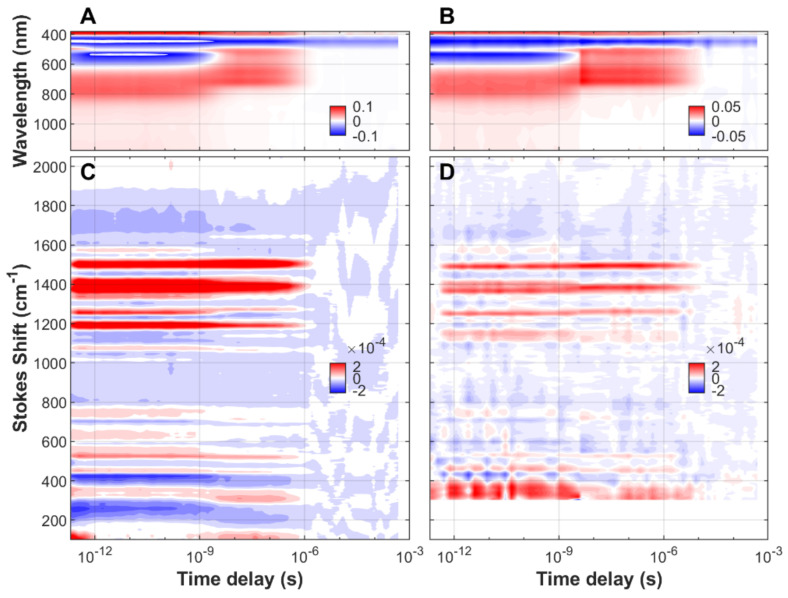
Time-resolved sub-picosecond to sub-millisecond Raman/absorption of FMN bound to EL222-WT upon excitation with a 475 nm actinic pump. (**A**) TA spectroscopy in H_2_O-based buffer. (**B**) TA spectroscopy in D_2_O-based buffer. (**C**) Transient FSRS in H_2_O-based buffer. (**D**) Transient FSRS in D_2_O-based buffer. Other parameters are as in Figure 1.

**Figure 3 biomolecules-13-00161-f003:**
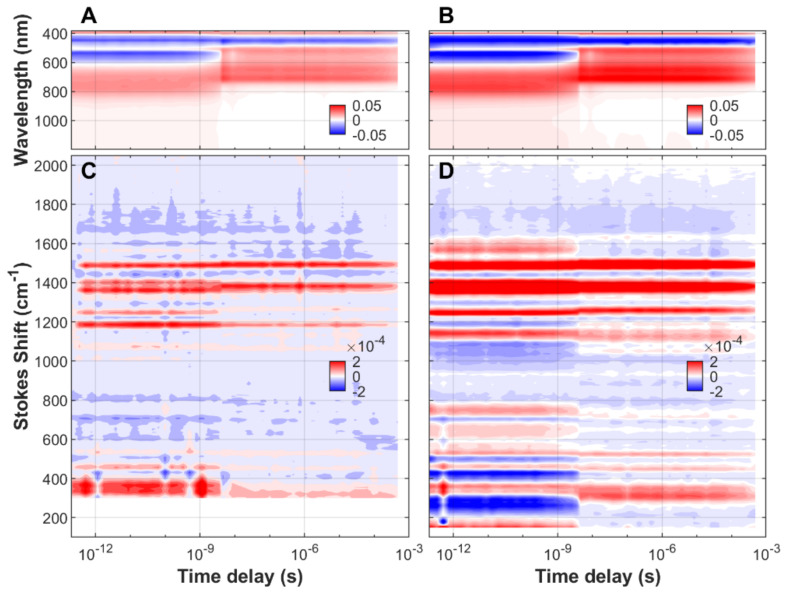
Time-resolved sub-picosecond to sub-millisecond Raman/absorption of FMN bound to EL222-C78A upon excitation with a 475 nm actinic pump. (**A**) TA spectroscopy in H_2_O-based buffer. (**B**) TA spectroscopy in D_2_O-based buffer. (**C**) Transient FSRS in H_2_O-based buffer. (**D**) Transient FSRS in D_2_O-based buffer. Other parameters are as in Figure 1.

**Figure 4 biomolecules-13-00161-f004:**
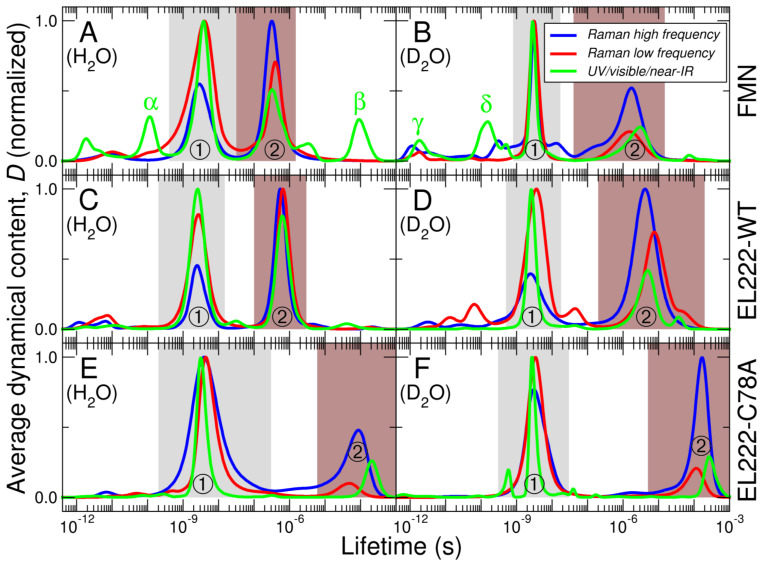
Normalized average dynamical content (*D*) as a function of lifetime. (**A**) FMN in H_2_O. (**B**) FMN in D_2_O. (**C**) EL222-WT in H_2_O. (**D**) EL222-WT in D_2_O. (**E**) EL222-C78A in H_2_O. (**F**) EL222-C78A in D_2_O. The curves have been calculated from the lifetime density plots shown in Appendix A, by averaging over three spectral regions: Raman high frequency (1750–1000 cm^−1^, blue lines), Raman low frequency (1000–303 cm^−1^, red lines), and UV/visible/near-IR (380–1000 nm, green lines). The two most abundant events present at all probed frequencies are labeled as ① (gray background) and ② (brown background). Extra dynamical events found at some but not all frequency ranges are labeled with Greek letters (α, β, γ, and δ).

**Table 1 biomolecules-13-00161-t001:** Mean lifetimes obtained from lifetime distribution analysis using the maximum entropy method. The units are indicated in the penultimate column (scale). A plausible interpretation for the observed lifetimes is indicated in the last column.

FMN						EL222-WT						EL222-C78A						Sample	
H_2_O			D_2_O			H_2_O			D_2_O			H_2_O			D_2_O			Buffer	
** RH **	** RL **	** TA **	** RH **	** RL **	** TA **	** RH **	** RL **	** TA **	** RH **	** RL **	** TA **	** RH **	** RL **	** TA **	** RH **	** RL **	** TA **	**Scale**	**Interpretation**
		115 (α)		1.7 (ε) 212 (ζ)	1.3 (γ) 152 (δ)													ps	Solvent [22] Ribityl [22]
2.9	3.3	3.8	3.2	1.9 (η) 3.1(θ)	2.9	2.5	2.7	2.6	2.5	3.7	2.6	3.9	4.4	3.2	3.1	3.4	2.7	ns	? ① ISC ①
0.32	0.40	0.32	1.7	1.5	3.0														?②
						0.55	0.66	0.63	4.2	7.6	4.9							µs	Adduct②
		91 (β)										87		208	109		269		Phospho- rescence or FMNH•? ②

RH = Raman high frequency region (1750–1000-cm^−1^, blue color). RL = Raman low frequency region (1000–149 cm^−1^ depending on the dataset, red color), TA = transient absorption (380–1000 nm, green color). ISC = intersystem crossing from singlet (1FMN*) to triplet (3FMN*). Adduct indicates 3FMN*-to-A390 transition. Phosphorescence equals to 3FMN*-to-S_0_ transition. FMNH• represents the neutral semiquinone radical species of FMN. The two most abundant events present at all probed frequencies are labeled as ① and ②. Unassigned events are labeled with a question mark. The adduct decay is beyond the experimental time window (~0.8 ms) and therefore could not be determined. Extra dynamical events found at some but not all frequency ranges are indicated with Greek letters (α, β, γ, δ, ε, ζ, η, and θ).

## Data Availability

Time-resolved transient absorption and FSRS datasets (Figure 1, Figure 2 and Figure 3) and their corresponding lifetime density maps (Appendix A) will be deposited in Zenodo: https://dx.doi.org/10.5281/zenodo.7389845.

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
