# Peer review of "Sub-Millisecond Photoinduced Dynamics of Free and EL222-Bound FMN by Stimulated Raman and Visible Absorption Spectroscopies"

_biomolecules, 2023, doi:10.3390/biom13010161_

Round 1
Reviewer 1 Report
The paper “Sub-millisecond Photoinduced Dynamics of Free and EL222-bound FMN by Stimulated Raman and Visible Absorption Spectroscopies” by Yingliang Liu et al. is a well written short communication regarding time-resolved FSRS/absorption of FMN and FMN bound to light activated DNA-binding protein EL222 and to a cysteine-less variant of EL222. The experimental set-up is able to record simultaneously transient absorption spectra in the UV/visible/near IR region and FSRS from femtoseconds to sub-milliseconds after photoexcitation, so spanning a wide wavelength and time range.
Nevertheless, a part of the results is not very clear (page 7, lines 229-244). Really it is not a results and discussion paragraph but, if the referee has understood well, the authors report the results obtained by other authors and compare them with theirs. The referee thinks that here discussion and results are interpenetrated, while they should be separated. In addition, the authors who obtained significant results in advance should be cited with their surname.
Page 7, lines 263-267. The authors observe in their spectra many other events in addition to the two explained in more detail and which are just known, but liquidate any interpretation writing that “A definitive assignment of the observed spectral bands linked to these extra dynamical events is beyond the scope of the present work.” The referee think that some hypothesis on possible assignments of the other events should be done.
Reviewer 2 Report
In the manuscript entitled Sub-millisecond Photoinduced Dynamics of Free and EL222-bound FMN by Stimulated Raman and Visible Absorption Spectroscopies, Liu et al. presented the study of EL222-FMN dynamics, explored via visTA/FSRS.
My opinion is that the manuscript is of acceptable quality as it is.
Author Response
We appreciate that the reviewer found the manuscript of sufficient quality.
Round 2
Reviewer 1 Report
The paper is now well argumented and do not require any other change.